# The critical role of natural forest as refugium for generalist species in oil palm-dominated landscapes

Sergio Guerrero-Sanchez[1,2]*, Benoit Goossens[1,2,3,4]*, Silvester Saimin[4], Pablo Orozco-terWengel[1]

**1** Organisms and Environment Division, School of Biosciences, Cardiff University, Cardiff, United Kingdom, **2** Danau Girang Field Centre, c/o Sabah Wildlife Department, Kota Kinabalu, Sabah, Malaysia, **3** Sustainable Places Research Institute, Cardiff University, Cardiff, United Kingdom, **4** Sabah Wildlife Department, Kota Kinabalu, Sabah, Malaysia

* ekio0474@gmail.com (SGS); GoossensBR@cardiff.ac.uk (BG)

**Data Availability Statement:** All relevant data are within the paper and the Supporting information files.

**Funding:** Guerrero-Sanchez was supported by the scholarship provided by the National Council for

## Abstract

In Borneo, oil palm plantations have replaced much of natural resources, where generalist species tend to be the principal beneficiaries, due to the abundant food provided by oil palm plantations. Here, we analyse the distribution of the Asian water monitor lizard (*Varanus salvator*) population within an oil palm-dominated landscape in the Kinabatangan floodplain, Malaysian Borneo. By using mark-recapture methods we estimated its population size, survival, and growth in forest and plantation habitats. We compared body measurements (*i.e.* body weight and body length) of individuals living in forest and oil palm habitats as proxy for the population's health status, and used general least squares estimation models to evaluate its response to highly fragmented landscapes in the absence of intensive hunting pressures. Contrary to previous studies, the abundance of lizards was higher in the forest than in oil palm plantations. Recruitment rates were also higher in the forest, suggesting that these areas may function as a source of new individuals into the landscape. While there were no morphometric differences among plantation sites, we found significant differences among forested areas, where larger lizards were found inhabiting forest adjacent to oil palm plantations. Although abundant in food resources, the limited availability of refugia in oil palm plantations may intensify intra-specific encounters and competition, altering the body size distribution in plantation populations, contrary to what happens in the forest. We conclude that large patches of forest, around and within oil palm plantations, are essential for the dynamics of the monitor lizard population in the Kinabatangan floodplain, as well as a potential source of individuals to the landscape. We recommend assessing this effect in other generalist species, as well as the impact on the prey communities, especially to reinforce the establishment of buffer zones and corridors as a conservation strategy within plantations.

Science and Technology (Consejo Nacional de Ciencia y Tecnología; CONACyT; scholarship No. 235294; Mexico Gov.). Fieldwork was supported by the Danau Girang Field Centre and Cardiff University through its PhD program.

**Competing interests:** The authors have declared that no competing interests exist

# Introduction

Anthropogenic habitat fragmentation has been pointed as one of the main drivers altering animal population dynamics, due to the reduction of suitable habitat and nutritional resources [1–3]. In Borneo, anthropogenic fragmentation has negatively affected populations of specialist species, such as the Bornean orang-utan (*Pongo pygmaeus morio*) [4], the Bornean sun bear (*Helarctos malayanus euryspillus*) [5, 6], the Bornean elephant (*Elephas maximus borneensis*) [7], and the Sunda clouded leopard (*Neofelis diardi*) [8], among others. On the other hand, these human-modified landscapes provide abundant food to generalist species, such as the Asian water monitor lizard (*Varanus salvator*) [9–11], the leopard cat (*Prionailurus bengalensis*) [12, 13], the bearded pig (*Sus barbatus*) [5, 14], the southern pig-tailed macaque (*M. nemestrina*) [15], and the Malay civet (*Viverra tangalunga*) [16]. Here, we implement a study to assess the population size and health status of the Asian water monitor lizard using its body size (*i.e.* body weight and body length [snout to vent length]) as a proxy to evaluate the status of the population and its response to highly fragmented landscapes in the absence of intensive hunting pressures.

The Asian water monitor lizard is one of the most successfully adapted species to anthropogenic habitats, despite facing intensive extraction rates throughout its distribution range [9, 11, 17]. In Borneo, the species is a top predator, just after the estuarine crocodile (*Crocodylus porosus*) and the Sunda clouded leopard, and it is linked to human-dominated habitats, where it is highly abundant [17, 18]. A few studies have suggested that the monitor lizard populations tend to be higher in areas of intense human activity such as villages [9, 11], and oil palm plantations, even in places where hunting is permitted [19, 20]. Twining et al. [19], for example, suggest that the abundance of lizards increases in a gradient of disturbance between natural forest and oil palm plantation. The species also has a high demand in the international pet and skin markets, as well as in the local market, particularly as a source of food and traditional medicine [21]. Although the Convention on International Trade in Endangered Species of Fauna and Flora (CITES) sets high extraction quotas for the Asian water monitor lizards in Malaysia [22], its current population status and the impact of human activities on the species remain unknown [21, 23]. The species is listed as "least concern" by the International Union for Conservation of Nature (IUCN) [24].

Earlier studies on the species' populations focused on determining the differences on abundance between natural forest and human-modified habitats using either distance surveys [9, 11] or mark-recapture methods [19, 20]. However, more detailed information is needed to properly evaluate the status of a population, especially when focusing on species that are target for either commercial or subsistence consumption [20, 25, 26]. Hence, survival and recruitment estimations, as well as body size distribution become fundamental parameters not only to accurately estimate extraction rates for commercial purposes, but also to assess the impact of landscape fragmentation on the health of the population. To our knowledge, there is no previous reports of the species population dynamics regarding recruitment and survival. However, for *V. komodensis*, it has been suggested that these parameters may be linked to the intensity of disturbance, as well as to island size and the distance among the study sites [27]. In regards to body size, although lizards inhabiting plantations with hunting pressure are reported to be smaller than in forested areas [20], other studies suggest that human-modified habitats, hold larger individuals than natural forest [11, 19]. Prey abundance and human-trophic subsidies are pointed to be the main driver of these size differences between forest and anthropogenic habitats.

The highly fragmented landscape of the Kinabatangan floodplain (Sabah, Malaysian Borneo), offers an opportunity to evaluate how oil palm-induced fragmentation impacts the

population of Asian water monitor lizards, in the absence of intensive hunting pressure. Since the 1950's, the Kinabatangan floodplain has experienced extensive deforestation, mainly due to commercial logging, and the conversion of almost half of the land into industrial oil palm plantations [28]. The expansion of oil palm along the Kinabatangan river is only limited by the Lower Kinabatangan Wildlife Sanctuary, established in 2005 by the State Government, in order to protect the remaining forest along the river [4]. Nonetheless, the connectivity between the sanctuary and the other forested areas (protected and non-protected) is either deficient, with narrow strips of highly degraded forest, or totally absent [28].

Our study aims to evaluate the influence of habitat fragmentation on the monitor lizard population and distribution within the landscape, through the estimation of its population size, body size distribution, and survival and growth rates. We expect to observe a positive influence of oil palm plantations on their population, as well as on the average of the individual body size, with higher densities and population growth rates in those anthropogenic habitats. We also expect the presence of larger animals in oil palm than in natural forest.

## Methods

### Ethics statement

Animal handling and sampling protocols were reviewed and authorized by Sabah Wildlife Department and the Sabah Biodiversity Centre, as part of the procedures to authorize access to natural resources, as well as for trapping and handling wild animals (permit number JKM/ MBS.1000-2/2 JLD.3-7).

### Study area

The Kinabatangan floodplain is the most productive wetlands in Sabah in terms of biomass [28, 29], with an annual rainfall of 3,000 mm and temperatures between 21–34˚C [7]. Within the floodplain, the Lower Kinabatangan Wildlife Sanctuary (LKWS) is a collection of 10 lots of degraded forest. It covers a total of 27,000 ha of protected forest along the Kinabatangan river, interconnected by small narrow corridors, and it is surrounded by large extensions of commercial oil palm plantations. The sanctuary is connected to seven protected areas, from the east coast to the central region of Sabah, as well as several patches of non-protected forest, which in total comprise approximately 75,000 ha of forest network with different grades of connectivity [4, 28]. Regardless of the degradation of its forest and its narrowness, the Sanctuary has been acknowledged as an important biodiversity stronghold in Sabah, as it is home to a unique assemblage of species of conservation priority in the region and some of their healthiest populations of priority species [4, 7, 28].

Different trapping sites were set in three forest lots of the LKWS (Lots 5, 6 and 7) and in three oil palm plantation estates (Hillco, Kopi and Kuril). Although the plantation sites within the study area seem homogeneous, they differ topographically; the terrain in the southern bank of the river (Kuril and Kopi plantation estates) varies in elevation and has an intricated network of limestone hills, while the plantation in the northern bank (Hillco) is completely flat. Most of the limestone hills in the plantations are indeed part of the cultivated area, however the intercalated valleys and slopes are permanently covered by ferns and dense riparian understory. On the contrary, in flat areas, the understory in the oil palm sites occurs in low densities or is totally absent, with exception of some small patches of swamp grasslands that are present in the area. Forest sites, on the other hand, only differ in their patch shape, as well as in the distance between trapping sites and the boundaries of the adjacent plantation. The expansion of oil palm in the area has left the Sanctuary as a narrow line of forest, which width varies from less than 50 m between the river and the boundary with the plantation, up to

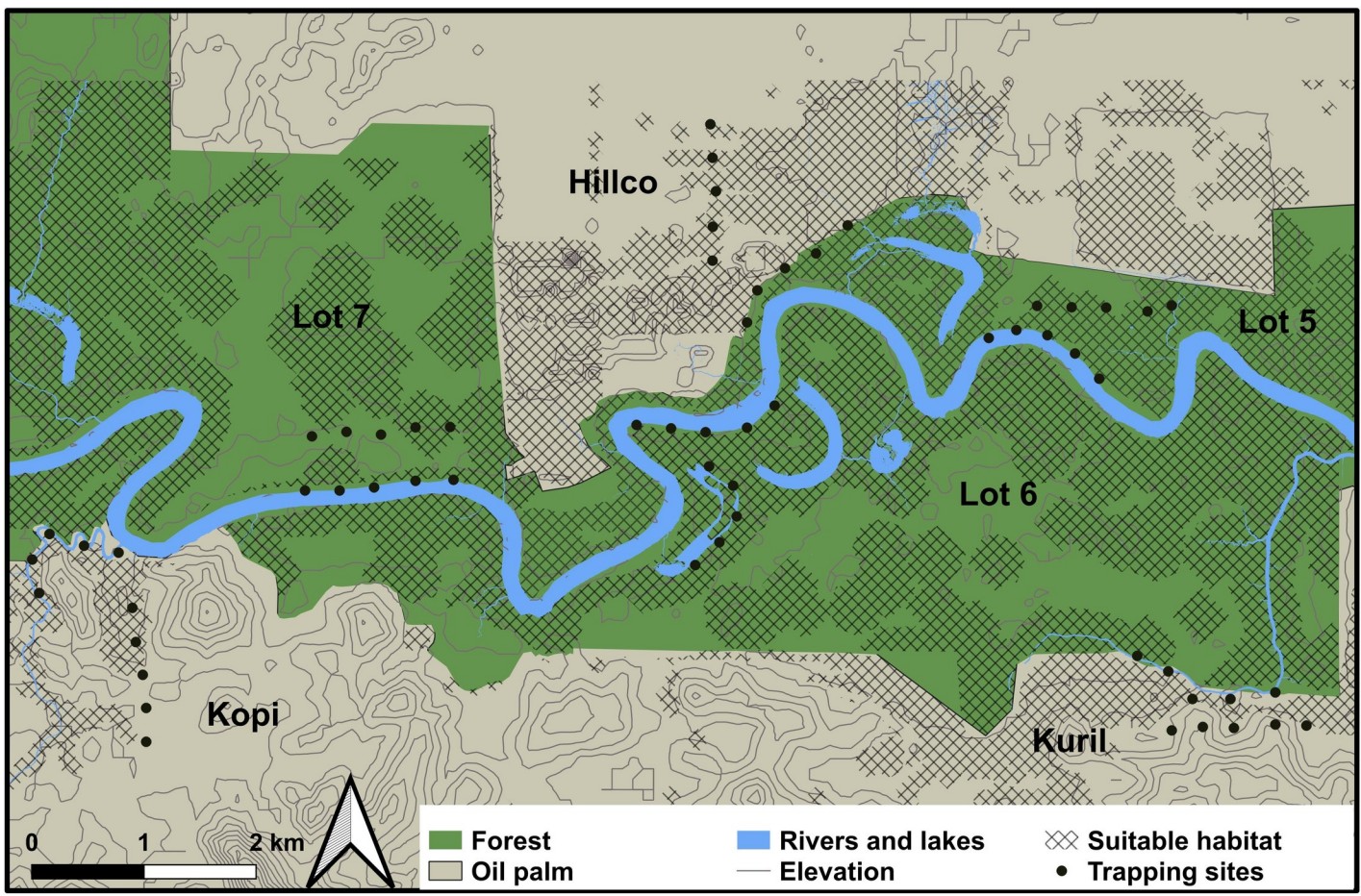

**Fig 1. Distribution of trapping sites and areas covered during the study in the Lower Kinabatangan Wildlife Sanctuary and surrounding plantations.**

around 1 km in the larger patches (lots). In regards to the study area, the trapping site set in Lot 5 was the closest to the plantation (<300 m), while in Lots 6 and 7, trapping sites were set farther from the boundaries of the adjacent plantation (>700 m; Fig 1).

In order to define the dimensions of the study area, we used GPS data generated by 14 successfully tagged individuals belonging to the same study group (unpublished). The limits of the study area were defined as the farthest points apart plus a buffer of 400 m based on home range estimations from the same study. Thus, the total study area corresponds to ~82.568 km$^2$, and it was divided into four study sites according to habitat type (forest and oil palm plantation) and location with respect to the Kinabatangan river (north and south; Fig 1).

## Trapping strategies

We used cage traps, which design was adapted from Purwandana et al. [27] and Ariefiandy et al. [30]. Traps were built out of wire mesh with the following dimensions: L = 90 cm, W = 40 cm, H = 40cm. Each trapping site consisted of two linear transects of ~1.6 km, where we deployed five cage traps every 400 m, along water streams (main river, tributaries or main drains), and in the interior (500 m inside forest or plantation). Due to the narrowed shape of the LKWS, finding habitat independence represents a significant challenge, especially when also looking for inter-transect independency. For the purpose of our study, we estimated that

setting the interior transects as far as 500 m from the river transects would be an adequate balance between feasibility and habitat independence, as well as inter-transect independence. However, although transects in both forest and oil palm areas were meant to be spatially standardized, traps in forest transects along the main river may differ regarding trapping success, with respect to traps in transects at the edge of the plantations. Such effects may reflect the size of water bodies present in the area (e.g. main river in forested areas v. tributary rivers and drains along the edge of plantation areas). While this issue was unavoidable due to the spatial conformation of the Lower Kinabatangan Wildlife Sanctuary and the size and shape of forest fragments remaining, such sampling design limitations should be taken into consideration when interpreting the results, and when replicating the study in different geographical areas. During trapping trials, different bait types were tested, *e.g.* food remains, fish and shrimps, chicken meat and chicken entrails, the latter being the most successful bait and thus, used in this study.

Traps were visited every morning (between 7:30 and 9:00 hrs), and revisited during the afternoon (between 15:00 and 17:00 hrs), right after the peak of activities of varanids and before their resting time [27, 31, 32]. Each trapped lizard was safely handled, measured, sampled and tagged with an intradermic transponder (ID-100A; Trovan Ltd., UK). The estimated area covered by each trap (*i.e.* trap influence area) was calculated as half the radius of the mean distance between all recapture sites [27, 32]. Each transect was sampled during 15 days every year (trapping season) for three consecutive years (October, 2013—September, 2016).

## Statistical analysis

**Population size, survival and growth rate.** Population size was estimated using the POPAN formulation for the Jolly-Seber method to estimate abundances for open populations [33–35]. This formulation is able to estimate the population size by considering apparent survival (Φ; hereafter **survival**), recapture probability (*p*), and probability of entry into the population at each occasion (*Pent*; hereafter **entry probability**). This method also considers variations between trapping seasons and between study groups, and assumes variable rates of incomes and outcomes, as well as marked individuals with homogeneous survival and catchability [34]. We used the program MARK v. 7.2 (Colorado State University; USA) to estimate population sizes with 95% confidence intervals, testing 93 different models that combined time and group variations for survival, recapture and entry probabilities. Population size (*N*) was considered with no temporal variations, but with variations among groups. The best model was selected by the lower corrected Akaike Information Criterion Index (AICc). The number of individuals estimated in the model would correspond to the sampling area, which is the area covered by the trap influence areas in each trapping site. Such numbers were extrapolated to a suitable area, defined as the area where the probability of presence of the species is relatively higher than the probability value in an average location in the training data (Probability Ratio Outcome; PRO > 1), with values larger than 1 representing the increase in likelihood of finding an animal (e.g. 2 corresponds to twice as likely) [36, 37]. Maximum Entropy (MAXENT) models for habitat selection with telemetry data (unpublished) were used to estimate the areas of higher probability of occurrence (PRO≥ 1). The same suitable area was also used to estimate the actual population density (ind./km$^2$).

Growth rate (λ) was calculated with Pradel's model using the same data set and considering 18 different combinations between the analysed variables. The model gives the growth rate estimation for each inter-season period. Such model differentiates survival from recruitment and considers them separately, avoiding over or underestimations [38]. Survival and recapture probabilities were calculated with variations and no variations over time and between groups,

while growth rate was calculated with variations between groups only. The outcome for the survival model is given as a probability factor in percentage, and the growth index is represented by a factor around the unit and can be understood either as increasing ($\lambda > 1.0$), decreasing ($\lambda < 1.0$) or stable ($\lambda = 1.0$) in time [38, 39]. The best fit model was selected as the model with the lowest AICc value.

**Body size distribution and effect on the population.** In order to understand how body size is distributed within the population, and how habitat may influence its distribution, we used Kruskal-Wallis test to assess differences regarding body weight and body length. The selected independent variables for comparison were habitat (forest and plantation) and study sites (three forest lots and three oil palm plantations). Analysis was carried out in the software R version 3.6.3 [40].

Additionally, we evaluated the association among population size, body weight and length, and suitable area for each trapping site, by using generalised least squares estimation models (GLS). These models deal with heteroscedasticity, and allow adjusting the variance structure of each variable, both continuous and discrete, to non-linear effect models [41]. We combined these structures using the *varFixed* function for the morphometric variables (continuous) and *varIdent* for the suitable area (discrete). Models were built separately for each habitat (forest and plantation). The analysis was carried out using the *nlme* v.3.1–152 package for R [42].

## Results

### Population size, survival and growth rates

A total of 3,055 day/traps were carried out with a 25% of recorded trapping success (774 capture events) and 402 individuals marked and sampled. During the whole period, no animals were recaptured in a different site other that its first capture. The best fit model for the population size estimation considered survival with no variation, neither over time nor among groups, entry probability with variations over time, recapture probability, and population size with only variations among groups (S1 Table in S1 File). The overall population size for the study area was estimated at 6,138 individuals (5,060–7,536) with an overall density of 127.49 ind./km$^2$ (105.09–156.51), however, we observed a larger number of individuals and higher densities in forested areas than in oil palm sites (Table 1). Regarding growth and survival rates, the best model assumed survival with no variations, and differences between groups for recapture probability and growth (S2 Table in S1 File). Both survival ($\Phi = 98.32 \pm 0.002$) and population growth ($\mu\lambda = 0.995–1$) suggest that the population is stable (Table 1).

**Table 1. Population size ($N$), survival ($\Phi$) and growth ($\lambda$) rates of the Asian water monitor lizard in the Kinabatangan floodplain.** Suitable area and adjusted $N$ were estimated according to suitable habitats for the species distribution.

| Study site | $N$ (min—max) | Sampling area (km$^2$) | Study area (km$^2$) | Suitable area (%*) | Adjusted N (min—max) | Density† (min—max) | $\Phi \pm$ SE | $\lambda \pm$ SE |
|---|---|---|---|---|---|---|---|---|
| Forest (north) | 725.54 (624–843) | 3.42 | 23.073 | 16.213 (70.27) | 3442 (2958–3996) | 212.3 (182.5–246.5) | 98.32 ± 0.002 | 0.996 ± 0.01 |
| Forest (south) | 199.45 (158–252) | 1.47 | 20.784 | 15.445 (74.31) | 2091 (1660–2684) | 135.4 (107.5–171.4) | | 0.999 ± 0.002 |
| Oil palm (north) | 78.27 (74–82) | 2.87 | 17.951 | 9.817 (54.69) | 267 (253–280) | 27.2 (25.8–28.6) | | 1 ± 0.002 |
| Oil palm (south) | 187.05 (104–337) | 3.68 | 23.700 | 6.673 (28.16) | 339 (189–611) | 50.8 (28.3–91.6) | | 0.995 ± 0.003 |
| Overall | | | | | 6138 (5060–7536) | 127.5 (105–156.5) | | |

* % of the study site's area.

† Density was estimated as ind./km$^2$.

## Recruitment

Recruitment (birth/immigration) estimations were overall higher in forested areas than in plantations in the two inter-season periods (ISP). We observed greater numbers in the northern forest (ISP-1 = 161.49 ± 16.61; ISP-2 = 117.25 ± 13.87), while Hillco, the northern oil palm estate showed the lowest recruitment rates (ISP-1 = 17.42 ± 1.45; ISP-2 = 12.65 ± 1.21): The southern study sites, however, showed similar recruitment rates among them, although values in forest are slightly higher than in oil palm. Our results suggest that forested areas may be more suitable as breeding sites, working as a source area of individuals for oil palm plantations. Nonetheless, they also show that plantations in the south bank may have suitable features (i.e. limestone hills with natural water streams and dense riparian understory in the slopes) to ensure the survival of new individuals (Table 2).

## Body weight and size distribution

Body weight calculated among trapped lizards ranged from 1.1 up to 25 kg, with an average of 5.86 ± 0.45 kg. The body length average was estimated in 63.91 ± 0.73 cm, ranging from 43.4 cm up to 107.80 cm. For both measurements, there were significant differences between forested areas and plantations. However, when we compared among sites in each habitat type separately, we only found significant differences for body weight in the forest (Kruskal-Wallis; $Xi^2$ = 8.33; df = 2; p = 0.015) and body length (Kruskal-Wallis; $Xi^2$ = 8.04; df = 2; p = 0.018), but not among oil palm plantations sites (Fig 2).

GLS models were tested to assess the effects of morphometric variables (body weight and length), and suitable area, on the population size in each habitat type. We observed that the size of suitable habitat in forested areas has a positive effect on the population size (GLS = 28.44 ± 6.62; t = 4.30; p < 0.001), but not in oil palm plantations. Instead, in these anthropogenic habitats, body length has a significant negative effect on the number of individuals in the population (GLS = -1.05 ± 0.49; t = -2.13; p = 0.034) (Table 3).

## Discussion

Although still a fragment of the whole picture, this study provides the most robust information so far about the population status of *V. salvator* in one of Borneo's fragmented landscapes. By using population density, growth rate, survival estimations, and body size distribution, we were able to provide a broader assessment of the influence of human-dominated landscapes on the population dynamics of this generalist carnivore. The population in the study area was highly abundant and stable, similar to what has been previously described for other populations of *V. salvator* in Indonesia (up to 1400 ind./km$^2$) [11], Peninsular Malaysia (37.9–372.4 ind./km$^2$) [20], and Sabah, Malaysian Borneo [19]. However, we observed that the number of individuals in oil palm plantations is not larger than that in forested areas, as previously suggested [19], indicating that natural habitats may have a fundamental role in the dynamics of this population at a landscape scale.

**Table 2. Recruitment estimation (birth/immigration) per inter-season period for each one of the four study sites within the Kinabatangan floodplain.**

| Inter-Season period | Birth/immigration estimates ± S.E. | | | |
|:---:|:---:|:---:|:---:|:---:|
| | Forest north | Forest south | Oil palm north | Oil palm south |
| 1 | 161.49 ± 16.61 (132–197) | 44.39 ± 6.27 (34–58) | 17.42 ± 1.45 (15–20) | 41.63 ± 13.03 (23–76) |
| 2 | 117.25 ± 13.78 (93–147) | 32.23 ± 4.72 (24–43) | 12.65 ± 1.21 (10–15) | 30.23 ± 9.71 (16–56) |

Values in parenthesis correspond to the range around the mean with a 95% C.I.

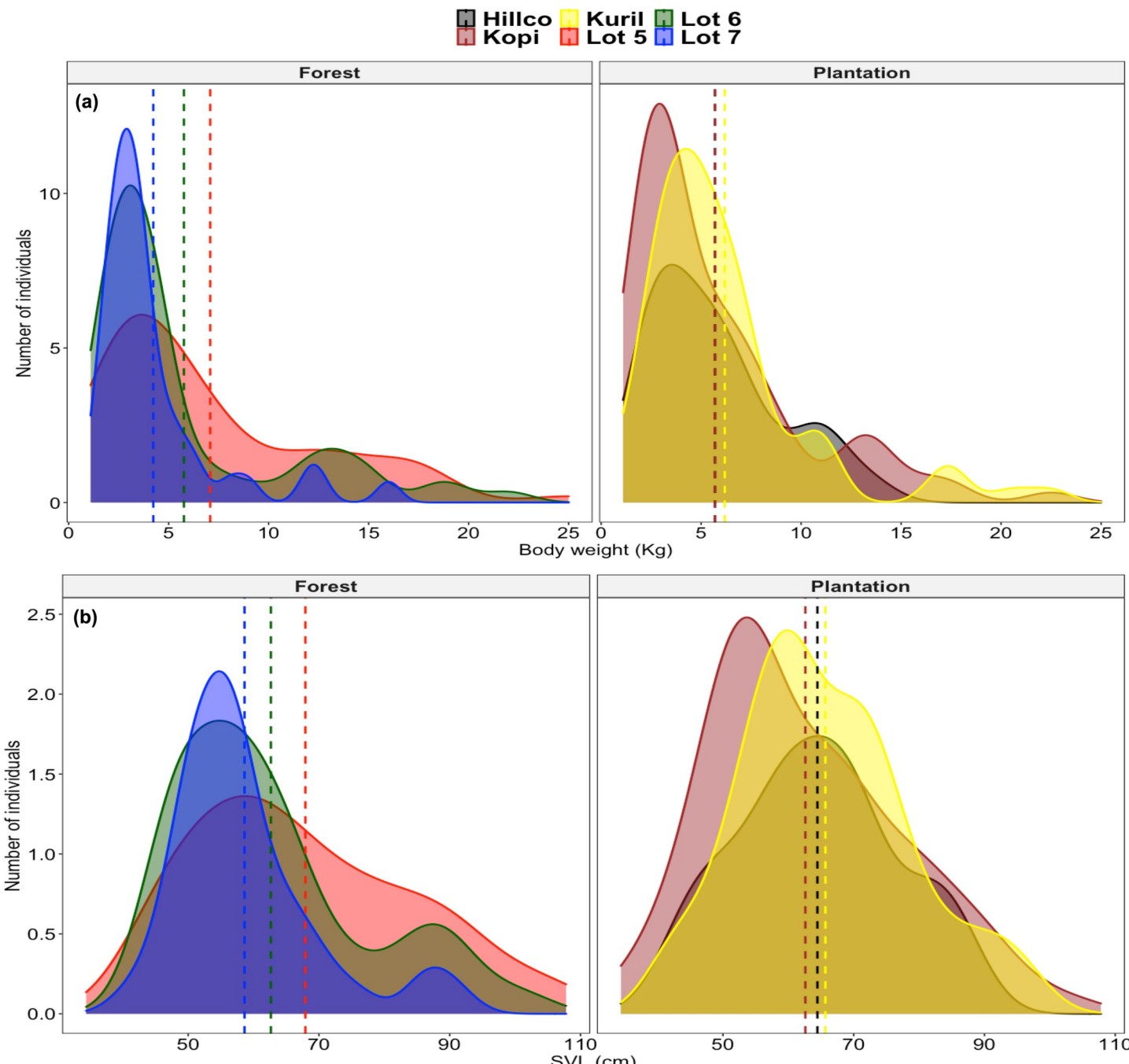

**Fig 2. Density plots for (a) body weight and (b) body length of the monitor lizard subpopulations per study site within the Kinabatangan floodplain.** Dashed lines show the mean value for each subpopulation.

The estimations obtained by Uyeda [11] could be either under- or over- estimated due to the methodology applied, as well as due to landscape features, and habitat disturbance by humans. Distance surveys have been shown to be less accurate for density estimations, compared with mark-recapture methods, especially when the target species is cryptic, and individual identification is difficult [30, 43, 44]. In Peninsular Malaysia, Khadiejah et al. [20] found

**Table 3. GLS model outcomes presenting the effect of body weight, body length, and the size of suitable areas, on the number of individuals estimated in each type of habitat.**

| Habitat | Variable | Value | S.E | t | P |
|---------|----------|-------|-----|---|---|
| Forest | Body weight | -13.353 | 21.604 | -0.618 | 0.537 |
| | Body length | 1.879 | 5.378 | 0.349 | 0.727 |
| | Suitable area | **28.437** | **6.167** | **4.297** | **<0.001** |
| Oil palm | Body weight | 1.871 | 2.253 | 0.830 | 0.407 |
| | Body length | **-1.049** | **-12.129** | **-2.129** | **0.034** |
| | Suitable area | 2.156 | 1.303 | 1.303 | 0.194 |

significantly higher abundances in mangroves and oil palm plantations exposed to intensive hunting pressures, than in rainforest habitats, where hunting was not permitted. It has been suggested that the abundance of food resources and the absence of other carnivore competitors may be the main drivers of high densities of monitor lizards in human-dominated habitats [11, 19, 20].

Our results, on the other hand, suggest that despite the abundance of food resources in oil palm plantations, the lack of protection in these habitats may have a negative effect on the number of individuals, especially when compared to neighbouring forest offering similar resources, in addition to a reduced number of carnivore competitors. In the LKWS, Sunda clouded leopard densities are estimated at 1.54 ind./100 km$^2$ [45], and, given the ecological niche of the estuarine crocodile, the species does not represent a competition for the Asian water monitor lizard [46, 47]. Hence, lizards may be able to thrive in these patches of forest as top predators, with little to no competition, but that of their own conspecifics.

The stability observed for the monitor lizard population in the Kinabatangan floodplain may be the result of landscape configuration and resource availability. However, oil palm plantations alone may not be able to offer enough suitable habitat for shelter, and high temperatures in these habitats, exacerbated by the lack of vegetation coverage, may be negatively impacting eggs and the survival of hatchlings [48–50]. The unsuitability of anthropogenic habitats would explain the lower number of individuals under 5 kg, as well as the lower recruitment rates in oil palm plantations than in forested areas.

Demographic similarities between Lot 6 and Kuril estate, in the southern bank of our study area, can be associated to intrinsic features of both sites. Although Lot 6 is a large patch of forest, it is frequently flooded; the drainage of accumulated water is slower and it has fewer shallow drains or streams than Lots 5 and 7 (*pers. obs.*). On the other hand, Kuril estate is an old plantation with an intricated network of hills covered by dense understory, rotten logs and small limestone caves that may be favourable for the survival of individuals inhabiting this site. While features in Lot 6 may have a negative impact on monitor lizard recruitment rates, Kuril estate may provide better protection than other plantation estates in the area. These findings suggest that, although forested areas seem to provide better refugia for monitor lizards, some environmental features in plantations may also be relevant for the survival of the population.

Our results suggest a sink-source-like dynamics, although not in a strict way, as described by Pulliam [51], where both mortality and natality rates differ in each site. Here, however, dispersal may play an important role in the dynamics of the population, keeping the balance between forest and oil palm plantation [52]. Although our results suggest that natural forests may have an important function as source areas of new individuals into the population, an inverse dynamic could be possible, especially in the south bank, where the topography and understory provide suitable environment for nests and shelter for hatchlings. The limited distribution of resources in oil palm, increases the chances of competition and antagonist

encounters with larger individuals, encouraging the new individuals to migrate into forested areas [52]. This critical functionality of forested areas in human-dominated landscapes has been described for other generalist species, such as the long-tailed weasel (*Mustela frenata*) [53], the roe deer (*Capreolus capreolus*) [54], and various murids [55, 56], where forested areas provide adequate protection and shelter, while the surrounding anthropogenic habitats supply abundant food.

Asian water monitor lizards have been reported to have larger body sizes when inhabiting anthropogenic habitats, compared to non-disturbed forest [11, 19]. However, in regions where hunting occurs, this pattern has been inverted [20]. As there is no record of intensive pressure on the species for commercial purposes in Sabah, hunting was not a variable under consideration for this study. Our results on body size differ from previous studies in that we did not find significant differences between lizards inhabiting oil palm plantations and those in forested areas. However, we did observe significant differences in body weight and body length among lizards living in different forest lots. Smaller lizards were more likely to be found in the two larger forest sites (Lots 6 and 7), while the number of large individuals was higher in Lot 5, where we observed a similar body size distribution to that in oil palm plantations. This similarity could be attributed to the size of this specific forest site and its proximity to a plantation estate (<300 m), providing lizards with abundant food and suitable protection, as well as a reduced chance of antagonistic encounters.

Density-dependent effects on population dynamics and natural history have been reported for different species [57], where both population and body size tend to be larger in the absence of predators, and in the presence of abundant food [58, 59]. The body size of Rosenberg's monitor (*V. rosenbergi)* in the Australian islands, for example, has been correlated with the abundance and the size of their prey species [58]. The results from the GLS models showed significant effect on the monitor lizard population size and the availability of suitable habitat in forested sites, but not in oil palm plantations. As oil palm plantations are characterized by an abundant presence of rodents [60], it is very likely that the high abundance of prey there may decrease that significance.

Moreover, the clustered distribution of these suitable areas in oil palm habitats, and their deficient connectivity, may increase intra-specific encounters and competition, influencing the body size of individuals, not as a result of food availability, but territoriality [61, 62]. Large monitor lizards are more tolerant to the presence of smaller lizards, than to similar sized or larger individuals [62]. Hence the significance of the negative association between body length and population size observed in oil palm habitats.

Contrastingly, the scattered distribution and connectivity of suitable areas in the forest, may reduce the chances of antagonistic intra-specific encounters, allowing individuals to thrive and develop in a less stressful environment. The broad distribution of resources in forested areas contributes to the coexistence of a larger number of individuals with no effects on their morphometrics. In the long term, the uneven distribution of the population within the landscape, due to the unbalanced distribution of resources may have consequences for the fitness of the population (*i.e.* nutrition, reproduction), as well as for the composition of prey communities in oil palm plantations and adjacent forested areas, especially along the boundaries [32, 53]. Our findings suggest that large patches of forest, as well as an increase in connectivity, could buffer these effects by allowing a better distribution and dispersion of the population within the landscape, and reducing intraspecific antagonism.

Finally, we highlight the importance of forested areas for generalist species living in oil palm-dominated landscapes. The structure and functional composition of the landscape matrix has a significant influence on the stability of the monitor lizard population in the Kinabatangan floodplain. As oil palm plantations have become conspicuous elements of Bornean

ecosystems, we strongly recommend evaluating their impact on other generalist carnivore species, as well as on prey communities. This information would go a long way towards reinforcing the importance of forest corridors and buffers, around and within plantation estates, but also within them, to allow for a healthy distribution and dispersal of wildlife populations, and to reduce the potential over-pressure on the resources provided by the surrounding forest.

## Supporting information

**S1 File.**
(PDF)

## Acknowledgments

We would like to thank the staff and students from the Danau Girang Field Centre for their help during the fieldwork, as well as the Director of Sabah Wildlife Department for his support, and Felda Global Holdings Sdn. Bhd. and Ladang Kinabatangan Sdn. Bhd. for granting permission to conduct part of the fieldwork within their plantation estates. The map in Fig 1 was generated by the authors with geographical information (forest and plantation layers) provided by Lucy Peter Liew, from the Danau Girang Field Centre. We also would like to thank our colleague, Liesbeth Frias, for her comments on early versions of this manuscript and proofreading.

## Author Contributions

**Conceptualization:** Sergio Guerrero-Sanchez, Benoit Goossens, Pablo Orozco-terWengel.

**Data curation:** Sergio Guerrero-Sanchez.

**Formal analysis:** Sergio Guerrero-Sanchez.

**Funding acquisition:** Benoit Goossens.

**Investigation:** Sergio Guerrero-Sanchez.

**Methodology:** Sergio Guerrero-Sanchez.

**Project administration:** Benoit Goossens.

**Resources:** Silvester Saimin.

**Supervision:** Benoit Goossens, Pablo Orozco-terWengel.

**Visualization:** Sergio Guerrero-Sanchez.

**Writing – original draft:** Sergio Guerrero-Sanchez, Benoit Goossens, Silvester Saimin, Pablo Orozco-terWengel.

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
