## [Decision Letter · Decision Letter 0]

11 Mar 2021

PONE-D-21-00412

The critical role of natural forest as refugium for generalist species in oil palm-dominated landscapes.

PLOS ONE

Dear Dr. Guerrero-Sanchez,

Thank you for submitting your manuscript to PLOS ONE. After careful consideration, we feel that it has merit but does not fully meet PLOS ONE’s publication criteria as it currently stands. Therefore, we invite you to submit a revised version of the manuscript that addresses the points raised during the review process.

We look forward to receiving your revised manuscript.

Kind regards,

Deborah Faria, PhD

Academic Editor

PLOS ONE

Additional Editor Comments:

Dear Dr. Guerrero-Sanchez,

I have received the reports from the advisors on your manuscript entitled " The critical role of natural forest as refugium for generalist species in oil palm-dominated landscapes ", which you submitted to the PlosOne.

Based on the reviews received, your manuscript could be considered for publication pending the incorporation of major revisions. For this reason, you are asked to carefully consider the comments of both reviewers. Both reviewers made an exceptional work, bringing sound suggestions while raised questions and concerns that should be tackled accordingly. I share the concerns of reviewer #2, regarding the possible river effect due to the spatial segregation of forests and plantations, and the statistical analysis in which you are using AIC values to compare models built from distinct data sets (different sampling sizes). These points need to be fully addressed in your review.

I am looking forward to receiving your revised manuscript within 30 days. Should you need more time to accomplish the revision please do not hesitate to inform me promptly.

Journal Requirements:

2. We note that Figure 1 in your submission contains map images which may be copyrighted.

We require you to either (a) present written permission from the copyright holder to publish this figure specifically under the CC BY 4.0 license, or (b) remove the figure from your submission:

b. If you are unable to obtain permission from the original copyright holder to publish this figure under the CC BY 4.0 license or if the copyright holder’s requirements are incompatible with the CC BY 4.0 license, please either i) remove the figure or ii) supply a replacement figure that complies with the CC BY 4.0 license. Please check copyright information on all replacement figures and update the figure caption with source information. If applicable, please specify in the figure caption text when a figure is similar but not identical to the original image and is therefore for illustrative purposes only.

Reviewers' comments:

Reviewer's Responses to Questions

**Comments to the Author**

1. Is the manuscript technically sound, and do the data support the conclusions?

Reviewer #1: Yes

Reviewer #2: Partly

2. Has the statistical analysis been performed appropriately and rigorously? 

Reviewer #1: Yes

Reviewer #2: No

3. Have the authors made all data underlying the findings in their manuscript fully available?

Reviewer #1: Yes

Reviewer #2: Yes

4. Is the manuscript presented in an intelligible fashion and written in standard English?

Reviewer #1: Yes

Reviewer #2: Yes

5. Review Comments to the Author

Reviewer #1: Dear authors,

this is an interesting piece of work on the effects of oil palm plantations and forests on the populations of a large monitor lizard. I understand that most theoretical ecologists would start criticizing your study areas: too few replicates and too heterogeneous environments (as one plantation is full of refuges due to the presence of limestone and one natural forest is susceptible to flooding, which may not represent a big problem for large monitors but maybe for offspring and eggs). However, I am a field biologist myself, and thus know how difficult it is sometimes to chose ideal areas in heavily altered environments. Sometimes forest fragments are so small that the whole fragment may "behave" as a forest edge. However, that should not impede the study of these areas because that could simply lead to nobody never studying them because they do not fit into an ideal sampling design.

As a whole I liked your study. It reads well and is far from being too wordy. Most parts have just the right size. However some issues need to be addressed before this manuscript could move to the next stage. Before I start I would like to mention that I am a frog specialist and thus maybe some of my observations do not perfectly fit for large lizards. So please feel free to refute those comments.

Lines 43-48: While your text dealing with conservation concerns including extraction quota sounds interesting it is irrelevant for your article as your aim is not to discuss these quotas or other threats that may be affecting this species. In fact it is enough to quote that it is highly abundant and linked to human-dominated habitats as you already do.

Lines 76-78: Your hypotheses need to be better explained and contextualized. If you write that you expect the presence of larger animals in anthropogenic habitats than in protected forests it would be nice to have some examples in the intro of species where something like that has already been observed.

Line 81-88: Something I am missing in the characterization of your study region is if the LKWS is achieving its mission, in other words, if it is really an effective conservation unit. Some conservation units may exist on paper but that is often no guarantee that they also exist in-situ as several studies have shown that conservation units, if not properly administered, suffer with logging, hunting and even with conversion to farmland. It would be good if you could explain a bit how effective this conservation unit is.

Line 92: You mention the limestone hills and how they turn the plantation more uneven. It would be interesting if you could write a bit more about these limestone hills: are they also covered by oil palms or are these tiny islands of other types of vegetation?

Line 101-106: Before reading your manuscript, just after reading the abstract I thought: Why did they only use mark recapture when tracking them with telemetry would have revealed so much additional and interesting data to answer their main questions? Tracking would allow to check if animals living on the borders (and in fact all your natural forest sites can be viewed as borders as the few hundred meters are no real obstacle for so large and fast moving monitors) use the plantation for feeding and the forests for shelter and kernels would also show how all this mosaic is used by them. Then I saw that at least you have data on tagged individuals. It is a pity that these data on tagged individuals has not already been published. It would have helped a lot to better understand this study here. Further, as the home range data has not been published it is impossible to check if they have been calculated accurately and using the best statistical approaches as this issue has been subject to very interesting discussions recently (see Fleming & Calabrese 2016 (https://doi.org/10.1111/2041-210X.12673). We have no other option than to trust that the unpublished manuscript has been done with the same accuracy and quality as the study we are reviewing here.

Line 110-115: I have some issues regarding your traps measuring 90x40x40 cm: Is this the standard device used to capture large monitors? If yes please cite some other studies where this traps were successfully used. Do these traps allow capture of small and large individuals with the same efficiency or are they more efficient for smaller individuals? I have trouble imagining a large monitor, e.g. of 107.80 cm (the largest from your own study) getting stuck in a 90x40x40 trap, but maybe it's a quite common event.

Line 189: While you state in your text that birth/immigration estimates are larger in forest areas your table 2 clearly shows that this is not valid for all studied areas as your southern forest area and your southern oil palm area show values that are very similar (44.39 vs 41.63). Your data only seems to be statistically significant when you roll all two forest areas and both oil palm areas into two areas.

Line 256-259: If I get you right natural forests may be playing a fundamental role as source areas of new monitors and oil palm plantations could be so called sinks. Maybe you could widen your discussion here including some literature on sources and sinks in populational biology (e. g. Dias 1996 (https://doi.org/10.1016/0169-5347(96)10037-9) or Kawecki 2004 (https://doi.org/10.1016/B978-012323448-3/50018-0) or Gravel et al. 2010 (https://doi.org/10.1890/09-0843.1) among others).

Line 260-267: That is exactly what is missing in your area description (see comment concerning line 92 above). Maybe what influences monitor densities is less related to an intact forest or not and more related to the presence of micro-niches and shelters. Kuril estate offers a lot of these refuges and thus functionally may be working as an old grown forest.

Line 286: "It" should not be capitalized.

Reviewer #2: Does the manuscript adhere to the PLOS Data Policy?

(Answer options: Yes, No) YES, IF THEY MAKE THEIR DATA AVAILABLE

This is an interesting and generally well-constructed and implemented study. However, I have two main concerns. The first is with the study design as the plots in forest appear to be mostly located along the river while the sites in oil palm are not. Therefore, the result you found could be nothing to do with forest versus oil palm but rather a river effect. I think you need to better describe the sampling design and how you paired forest and palm oil sites to avoid these types of confounding effects. And if there are potential confounding effects they need to, at the very least, be controlled for in the statistical analysis. The second issue is that you appear to compare the AIC from models fitted to different data sets. You cannot do this as the comparison is confounded by the effect of sample size and data on the likelihood. I expand on these comments below and also include a number of other comments.

Lines 41-43: The species is highly abundant but you also state it is threatened – these seem to be contradictory.

Lines 48-54: Distance sampling and mark-recapture actually provide different types of information so I think it is less about accuracy and more about the purpose of the survey.

Sampling design. Need more information about the sampling design and how you matched plantation and forest plots to control for confounding variables (for example, why are all the forest plots along the river? This is likely to bias your results).

Statistical analysis. What is the spatial unit for which you estimated population size? Is it the transect? This should be made clear.

Lines 149-157. What do you mean by “suitable area” and how did you measure it?

Lines 161-162. How many recaptures where there in total? I presume here you mean that an individual trapped at one site was never recaptured at another site rather than there were no recaptures at all?

I suggest avoid using acronyms as the manuscript is difficult to read with the use of acronyms so often (I had to keep reminding myself what each acronym meant).

Lines 200-204. You cannot compare the AIC of models fitted to different data sets (which seems to be what you are doing here unless I am mistaken).

Lines 205-207. I wonder whether this is because you measured the size of habitat differently between the two habitat types. It is hard to tell since you don’t describe how you measured the size of habitat.

Table 3. See my comments above about not comparing AIC among models with different data sets. The AIC for the full data set will have a larger AIC since it has more data points so the likelihood is larger (hence cannot be compared to the models fitted to the other data sets).

Line 230. less accurate that what other methods?

Line 275. Do you mean “density dependent”?

6. PLOS authors have the option to publish the peer review history of their article (what does this mean?). If published, this will include your full peer review and any attached files.

Reviewer #1: **Yes: **Mirco Solé

Reviewer #2: **Yes: **JONATHAN RHODES

---

## [Author Response · Author response to Decision Letter 0]

27 Apr 2021

Dear editor and reviewers:

We appreciate your valuable comments to our manuscript. We have considered each of them carefully and responded accordingly. A document of response to each reviewer has been attached, along with a revised manuscript with tracking changes, as well as a clean version of the manuscript.

In regards to figure 1, the map was produced by the first author (Guerrero-Sanchez) with geographic information (forest and plantation layers) provided by the Danau Girang Field Centre, we have included a mention to the officer, Lucy P. Liew in the acknowledgements. The trapping sites and “suitable habitat” was generated by Guerrero-Sanchez with information generated by himself (we have attached a response document for this matter).

We hope this help to avoid confusions on copyright.

In response to the comments from reviewer 2, we have made clearer both our sampling design and the statistical analysis. Both clarifications are addressed in our response to the reviewer.

---

## [Decision Letter · Decision Letter 1]

20 Jul 2021

PONE-D-21-00412R1

The critical role of natural forest as refugium for generalist species in oil palm-dominated landscapes.

PLOS ONE

Dear Dr. Guerrero-Sanchez,

Thank you for submitting your manuscript to PLOS ONE. After careful consideration, we feel that it has merit but does not fully meet PLOS ONE’s publication criteria as it currently stands. Therefore, we invite you to submit a revised version of the manuscript that addresses the points raised during the review process.

We look forward to receiving your revised manuscript.

Kind regards,

Bilal Habib

Academic Editor

PLOS ONE

Journal Requirements:

Additional Editor Comments (if provided):

I am happy with the revision, but one of the reviewers have still some issues with resect to methodology. If you are unable to answer the questions with resect to concerns raised by reviewer, i suggest adding paragraph about the limitations of the study and highlighting this particular issue for further research. I hope this is helpful.

Reviewers' comments:

Reviewer's Responses to Questions

**Comments to the Author**

1. If the authors have adequately addressed your comments raised in a previous round of review and you feel that this manuscript is now acceptable for publication, you may indicate that here to bypass the “Comments to the Author” section, enter your conflict of interest statement in the “Confidential to Editor” section, and submit your "Accept" recommendation.

Reviewer #1: All comments have been addressed

Reviewer #2: (No Response)

2. Is the manuscript technically sound, and do the data support the conclusions?

Reviewer #1: Yes

Reviewer #2: No

3. Has the statistical analysis been performed appropriately and rigorously? 

Reviewer #1: Yes

Reviewer #2: No

4. Have the authors made all data underlying the findings in their manuscript fully available?

Reviewer #1: Yes

Reviewer #2: No

5. Is the manuscript presented in an intelligible fashion and written in standard English?

Reviewer #1: Yes

Reviewer #2: Yes

6. Review Comments to the Author

Reviewer #1: Dear authors

I am glad that the comments made by both reviewers have helped to turn your manuscript even better than it already was. You have addressed all comments in a very positive and constructive way and thus I feel that your study is now ready for the next stage.

Reviewer #2: Although you have partially addressed my previous concerns, I feel some still remain unaddressed in your responses. My remaining concerns are as follows:

"R2-C2. Lines 48-54: Distance sampling and mark-recapture actually provide different types of

information so I think it is less about accuracy and more about the purpose of the survey."

In your response you state that Distance Sampling cannot account for variation in detectability between the two habitats, but this is strictly untrue. This is exactly what Distance Sampling is designed to do and I suggest you modify this statement.

"R2-C3. Sampling design. Need more information about the sampling design and how you

matched plantation and forest plots to control for confounding variables (for example, why are all the forest plots along the river? This is likely to bias your results)."

In your response you seem to explain how you achieved independence among sampling sites not how you dealt with confounding factors. I understand that accounting for confounding factors may be difficulty, but at the very least it needs to be discussed as a major limitation (if they can’t be dealt with) and provide an explanation that, if there are unaccounted for confounding factors, the ability to generalise your results to other systems will be limited (i.e., making general statements about the differences between oil palm and natural forest will not be possible).

"R2-C4. Statistical analysis. What is the spatial unit for which you estimated population size? Is it the transect? This should be made clear."

In your modified paragraph what does “higher probability of occurrence” really mean? I think you need to be much more specific here. I assume you set a threshold that defined suitable habitat and if this is what you did then that threshold needs to be defined.

"R2-C8. Lines 200-204. You cannot compare the AIC of models fitted to different data sets (which seems to be what you are doing here unless I am mistaken)."

In your response you never answer my question about whether you compare AIC values for models fitted to different data sets. It still sounds like you did, which would be incorrect if so.

Overall I feel that the questions I raised still need some more careful thinking about and more complete and targeted responses.

Jonathan Rhodes

7. PLOS authors have the option to publish the peer review history of their article (what does this mean?). If published, this will include your full peer review and any attached files.

Reviewer #1: **Yes: **Mirco Solé

Reviewer #2: **Yes: **Jonathan Rhodes

---

## [Author Response · Author response to Decision Letter 1]

10 Aug 2021

Responses to Reviewer 2:

Although you have partially addressed my previous concerns, I feel some still remain unaddressed in your responses. My remaining concerns are as follows:

"R2-C2. Lines 48-54: Distance sampling and mark-recapture actually provide different types of information so I think it is less about accuracy and more about the purpose of the survey."

In your response you state that Distance Sampling cannot account for variation in detectability between the two habitats, but this is strictly untrue. This is exactly what Distance Sampling is designed to do and I suggest you modify this statement.

RE: The paragraph has been modified as follows:

“Earlier studies on the species’ populations focused on determining the differences on abundance between natural forest and human-modified habitats using either distance surveys or mark-recapture methods. However, a more detailed information is needed to properly evaluate the status of a population, especially when focusing on species that are target for either commercial or subsistence consumption” (Actual lines 55-59).

"R2-C3. Sampling design. Need more information about the sampling design and how you

matched plantation and forest plots to control for confounding variables (for example, why are all the forest plots along the river? This is likely to bias your results)."

In your response you seem to explain how you achieved independence among sampling sites not how you dealt with confounding factors. I understand that accounting for confounding factors may be difficulty, but at the very least it needs to be discussed as a major limitation (if they can’t be dealt with) and provide an explanation that, if there are unaccounted for confounding factors, the ability to generalise your results to other systems will be limited (i.e., making general statements about the differences between oil palm and natural forest will not be possible).

RE: We added a paragraph in the methods section acknowledging the limitations of the sampling method and we recommended to take the outcomes of the study with caution:

“However, although transects in both forest and oil palm areas were meant to be spatially standardized, traps in forest transects along the main river may differ regarding trapping success, with respect to traps in transects at the edge of the plantations. Such effects may reflect the size of water bodies present in the area (e.g. main river in forested areas v. tributary rivers and drains along the edge of plantation areas). While this issue was unavoidable due to the spatial conformation of the Lower Kinabatangan Wildlife Sanctuary and the size and shape of forest fragments remaining, such sampling design limitations should be taken into consideration when interpreting the results, and when replicating the study in different geographical areas.” (Actual lines 143-151).

"R2-C4. Statistical analysis. What is the spatial unit for which you estimated population size? Is it the transect? This should be made clear."

In your modified paragraph what does “higher probability of occurrence” really mean? I think you need to be much more specific here. I assume you set a threshold that defined suitable habitat and if this is what you did then that threshold needs to be defined.

RE: We defined the area with “higher probability of occurrence” as the area generated by the MAXENT model with a probability ratio of occurrence above the unit (PRO ≥ 1). It is the output created by the predictive model where 1 represents the probability of occurrence in an “average” location in the training data, while any X value above the unit represent X times higher than the average probability of presence (e.g. a 2 would imply a twice as high probability of presence). The paragraph was modified as follows:

“The number of individuals estimated in the model would correspond to the sampling area, which is the area covered by the trap influence areas in each trapping site. Such numbers were extrapolated to a suitable area, defined as the area where the probability of presence of the species is relatively higher than the probability value in an average location in the training data (Probability Ratio Outcome; PRO > 1), with values larger than 1 representing the increase in likelihood of finding an animal (e.g. 2 corresponds to twice as likely). Maximum Entropy (MAXENT) models for habitat selection with telemetry data (unpublished) were used to estimate the areas of higher probability of occurrence (PRO≥ 1). The same suitable area was also used to estimate the actual population density (ind./km2).” (Actual lines 177-186).

"R2-C8. Lines 200-204. You cannot compare the AIC of models fitted to different data sets (which seems to be what you are doing here unless I am mistaken)."

In your response you never answer my question about whether you compare AIC values for models fitted to different data sets. It still sounds like you did, which would be incorrect if so.

Re: We have modified the respective paragraph as follows:

“GLS models were tested to assess the effects of morphometric variables (body weight and length), and suitable area, on the population size in each habitat type. We observed that the size of suitable habitat in forested areas has a positive effect on the population size (GLS = 28.44 ± 6.62; t = 4.30; p < 0.001), but not in oil palm plantations. Instead, in these anthropogenic habitats, body length has a significant negative effect on the number of individuals in the population (GLS = -1.05 ± 0.49; t = -2.13; p = 0.034)”. (Actual lines 258-263; Table 3 has been modified too).

---

## [Editor Report · Decision Letter 2]

13 Sep 2021

The critical role of natural forest as refugium for generalist species in oil palm-dominated landscapes.

PONE-D-21-00412R2

Dear Dr. Guerrero-Sanchez,

We’re pleased to inform you that your manuscript has been judged scientifically suitable for publication and will be formally accepted for publication once it meets all outstanding technical requirements.

Kind regards,

Bilal Habib

Academic Editor

PLOS ONE

Additional Editor Comments (optional):

The responses to the comments are satisfactory, the paper shall be accepted for publications.
---

## [Editor Report · Acceptance letter]

15 Sep 2021

PONE-D-21-00412R2 

The critical role of natural forest as refugium for generalist species in oil palm-dominated landscapes. 

Dear Dr. Guerrero-Sanchez:

I'm pleased to inform you that your manuscript has been deemed suitable for publication in PLOS ONE. Congratulations! Your manuscript is now with our production department. 

Kind regards, 

on behalf of

Dr. Bilal Habib 

Academic Editor

PLOS ONE